# Geometric Order Learning for Rank Estimation

**Seon-Ho Lee**                **Nyeong-Ho Shin**                **Chang-Su Kim**

School of Electrical Engineering
Korea University
seonholee@mcl.korea.ac.kr, nhshin@mcl.korea.ac.kr, changsukim@korea.ac.kr

## Abstract

A novel approach to rank estimation, called geometric order learning (GOL), is proposed in this paper. First, we construct an embedding space, in which the direction and distance between objects represent order and metric relations between their ranks, by enforcing two geometric constraints: the order constraint compels objects to be sorted according to their ranks, while the metric constraint makes the distance between objects reflect their rank difference. Then, we perform the simple $k$ nearest neighbor ($k$-NN) search in the embedding space to estimate the rank of a test object. Moreover, to assess the quality of embedding spaces for rank estimation, we propose a metric called discriminative ratio for ranking (DRR). Extensive experiments on facial age estimation, historical color image (HCI) classification, and aesthetic score regression demonstrate that GOL constructs effective embedding spaces and thus yields excellent rank estimation performances. The source codes are available at https://github.com/seon92/GOL

## 1 Introduction

In rank estimation, we estimate the rank (or ordered class) of an object. It is different from ordinary classification, for its classes are arranged in a natural order. For example, in movie rating, classes can be ordered from 'outstanding' to 'very good,' 'satisfactory,' 'unsatisfactory,' and 'poor.' Rank estimation is a fundamental problem and, *e.g.*, used for various computer vision tasks including facial age estimation (Shin et al., 2022), aesthetic quality assessment (Schifanella et al., 2015), and HCI classification (Palermo et al., 2012).

For rank estimation, many techniques (Li et al., 2021; Liu et al., 2018) adopt the ordinal regression framework, which employs a classifier or a regressor to predict the rank of an object directly. However, they may fail to yield reliable estimates, for there is no clear distinction between ranks in many cases. For instance, in facial age estimation, the aging process — causing variations in facial shapes, sizes, and texture — has large individual differences due to factors such as genes, diet, and lifestyle.

To address this issue, comparison-based algorithms (Lim et al., 2020; Lee & Kim, 2021; Li et al., 2014; Nguyen et al., 2018) have been proposed. Instead of predicting the rank directly, they learn a binary relation between objects, such as order or metric (Hrbacek & Jech, 1984). These relations provide useful information for rank estimation: an order indicates the relative priority between objects $x$ and $y$, while a metric informs of the distance between them.

In order learning (Lim et al., 2020; Lee & Kim, 2021), a comparator is learned to classify the relationship between $x$ and $y$ into one of three cases: $x$ is 'greater than,' 'similar to,' or 'smaller than' $y$. Then, they estimate the rank of a test object by comparing it with multiple reference objects with known ranks. This approach is based on the idea that it is easier to predict ordering relationships between objects than to estimate the absolute ranks; telling the older one between two people is easier than estimating their exact ages. However, order learning disregards how much $x$ is different from $y$. In other words, it ignores metric information.

36th Conference on Neural Information Processing Systems (NeurIPS 2022).

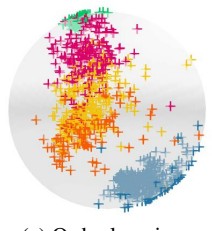 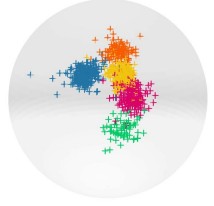 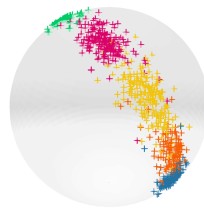

(a) Order learning        (b) Metric learning        (c) Proposed

Figure 1: Comparison of embedding spaces of (a) order learning (Lim et al., 2020), (b) metric learning (Schroff et al., 2015), and (c) the proposed algorithm. For each method, we train ResNet18 (He et al., 2016) to classify the RetinaMNIST data (Yang et al., 2021) into the five ordinal classes, depicted by blue, orange, yellow, pink, and green markers. For visualization on the unit sphere, the output dimension of a network is reduced to 3 via a fully connected layer, and each feature is normalized.

On the other hand, metric learning algorithms (Li et al., 2014; Nguyen et al., 2018) employ the triplet constraint on three objects $(x, y, z)$. It enforces the distance between $x$ and $y$ to be less than that between $x$ and $z$ in the embedding space if the ranks of $(x, y, z)$ are in the increasing or decreasing order. By its design, the triplet constraint does not fully exploit the order among objects.

Figures 1 (a) and (b) compare embedding spaces, obtained by order learning and metric learning, respectively. Order learning sorts instances according to ranks in general, but instances in each class are scattered in the embedding space. In contrast, metric learning reduces within-class scattering but does not sort instances properly. For example, both blue and orange ranks are adjacent to the yellow one, although they should be arranged in the order of blue, orange, and yellow. Unlike ordinary classification, different errors have different severities in rank estimation: misclassifying an object in the blue rank as a yellow one is severer than mistaking it for an orange one. Ordering in the embedding space is important to avoid such severe errors.

We propose a novel algorithm, GOL, to estimate the rank of an object reliably by exploiting both order and metric relations. To this end, we construct an embedding space, in which the direction and distance between objects represent the order and metric relations between their ranks. For the construction, we formulate two geometric constraints in the embedding space: 1) the order constraint enforces the feature vectors of instances to be arranged according to their ranks, and 2) the metric constraint makes the distance between instances reflect their rank difference. To satisfy these two constraints simultaneously, we introduce reference points that guide the region of each rank in the embedding space. Then, we use the simple $k$-NN rule in the embedding space to estimate the rank of a test instance. Extensive experiments show that GOL constructs high-quality embedding spaces and thus provides excellent rank estimation performances.

The contributions of this paper can be summarized as follows.

- GOL is the first attempt to design an embedding space in which the direction and distance between objects represent their order and metric relations.
- We introduce a novel metric, called DRR, to assess the quality of embedding spaces for rank estimation. Then, it is shown that GOL effectively sorts and separates instances according to their ranks in an embedding space, as illustrated in Figure 1 (c).
- GOL achieves state-of-the-art performances on various benchmark datasets for facial age estimation, HCI classification, and aesthetic score regression. Specifically, GOL performs the best in 20 out of 25 benchmark tests.

## 2    Related Work

**Ordinal regression:** Many ordinal regression methods have been developed to estimate the rank of an object directly using classifiers or regressors. Rothe et al. (2015) employed tens of classifiers to yield the average of their predictions as output. Yi et al. (2014) developed a regressor to estimate the rank of an image using multi-scale patches. Also, Frank & Hall (2001) employed multiple binary classifiers, each of which tells whether the rank of an object is higher than a series of thresholds or not. However, such direct estimation of ranks is challenging even for human beings in general; *e.g.*,

humans usually predict only a rough range of another one's age with limited confidence. Thus, Diaz & Marathe (2019) trained a regressor using soft ordinal labels to alleviate penalties on close predictions. Furthermore, Li et al. (2021) and Li et al. (2022) modeled the uncertainty of each prediction as a Gaussian distribution. In contrast to these methods, the proposed algorithm provides more accurate rank estimates, although it uses only a single encoder network with the simple $k$-NN rule and makes no complicated probabilistic assumptions.

**Order learning:** Lim et al. (2020) first proposed the notion of order learning, which learns ordering relationships between objects and determines the rank of an unseen object by comparing it with references with known ranks. It yields promising results because relative assessment is easier than absolute assessment in general. Lee & Kim (2021) improved the performance of order learning by finding more reliable references. They decomposed object information into an order-related feature and an identity feature and showed that objects with similar identity features can be compared more reliably. Also, Shin et al. (2022) extended the classification approach in (Lim et al., 2020; Lee & Kim, 2021) to a regression-based one. These order learning methods, however, require a significant computational cost to find reliable references from an entire training set. Moreover, for rank estimation, they should do comparisons with many references with different ranks because they consider only relative priorities between objects. On the contrary, the proposed algorithm simply carries out the $k$-NN search to yield outstanding rank estimation results.

**Metric learning:** Metric learning aims to construct an embedding space in which the distance between objects reflects their semantic difference. Most metric learning algorithms (Schroff et al., 2015; Deng et al., 2019) are for ordinary classification, clustering, or image retrieval tasks. Therefore, they enforce an object to be located near other objects in the same class in the embedding space but far from objects in different classes. However, in rank estimation, this approach may be suboptimal because it does not consider ordinal relationships among classes. For example, in movie rating, it does not discriminate the class difference between 'outstanding' and 'very good' from that between 'outstanding' and 'poor.' To alleviate this problem, Xiao et al. (2009) designed a metric, called labeled distance, to measure semantic similarities between objects and attempted to preserve local semantic structures in the feature space. Moreover, to preserve the ordinal relationships, Li et al. (2012) developed a metric learning algorithm to make the distances between objects proportional to their rank differences. Also, Tian et al. (2016) employed a series of margins to explicitly impose different embedding distances according to rank differences. Suárez et al. (2021) attempted to sort the embedding distances between pairs of objects according to their rank differences.

## 3 Proposed Algorithm

### 3.1 Preliminary – Order and Metric

Mathematically, both order and metric are binary relations (Hrbacek & Jech, 1984). An *order* (Schröder, 2003), denoted by $\leq$, on a set $\Theta = \{\theta_0, \theta_1, \ldots, \theta_{M-1}\}$ should satisfy the properties of

- Reflexivity: $\theta_i \leq \theta_i$ for all $i$,
- Antisymmetry: $\theta_i \leq \theta_j$ and $\theta_j \leq \theta_i$ imply $\theta_i = \theta_j$,
- Transitivity: $\theta_i \leq \theta_j$ and $\theta_j \leq \theta_k$ imply $\theta_i \leq \theta_k$.

On the other hand, a *metric* (Rudin, 1991) is a distance function $d$ satisfying

- Nonnegativity: $d(\theta_i, \theta_j) \geq 0$ for all $i, j$, and $d(\theta_i, \theta_j) = 0$ if and only if $\theta_i = \theta_j$,
- Commutativity: $d(\theta_i, \theta_j) = d(\theta_j, \theta_i)$ for all $i, j$,
- Triangle inequality: $d(\theta_i, \theta_k) \leq d(\theta_i, \theta_j) + d(\theta_j, \theta_k)$ for all $i, j, k$.

In rank estimation, an order describes the priorities of ranks or classes in the set $\Theta = \{\theta_0, \ldots, \theta_{M-1}\}$, where each rank represents one or more object instances. For example, in age estimation, $\theta_i$ may represent $i$-year-olds, and $\theta_{17} < \theta_{32}$ indicates that 17-year-olds are younger than 32-year-olds. Let $\theta(\cdot)$ be the rank function, and let $x$ and $y$ be instances. Then, $\theta(x) = \theta_{17}$ means that person $x$ is 17-year-old. Also, a metric describes the difference between ranks in $\Theta$. For example, $d(\theta(x), \theta(y)) = 15$ means that two people $x$ and $y$ are 15 years apart.

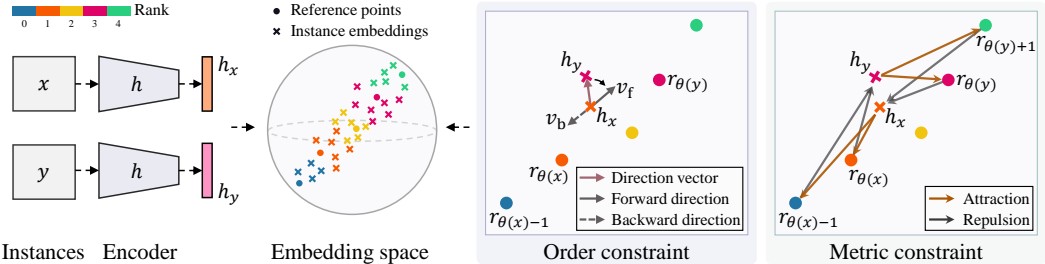

Figure 2: An overview of the proposed GOL algorithm. It is recommended to watch the accompanying video of the proposed algorithm.

## 3.2 Embedding Space Construction

Given an object instance $x$, the objective is to estimate its rank $\theta(x)$. In such rank estimation, an order and a metric convey complementary information: the order provides directional information between ranks, while the metric does length (or magnitude of difference) information. In age estimation, let us consider three age ranks $\theta_2$, $\theta_{17}$, and $\theta_{32}$. Since $\theta_2 < \theta_{17} < \theta_{32}$, the order informs that $\theta_2$ and $\theta_{32}$ are at the opposite sides with respect to $\theta_{17}$. On the other hand, since $d(\theta_2, \theta_{17}) = d(\theta_{32}, \theta_{17}) = 15$, the metric indicates how far $\theta_2$ and $\theta_{32}$ are from $\theta_{17}$. In this case, both lengths are identically 15.

For rank estimation, the pairwise comparison methods (Lim et al., 2020; Lee & Kim, 2021; Nguyen et al., 2018) attempt to learn these relations. However, Lim et al. (2020) and Lee & Kim (2021) exploit the order relation only, whereas Nguyen et al. (2018) use the metric relation only to train their neural networks. Thus, the conventional methods may yield sub-optimal results.

To learn both order and metric relations, we propose a geometric approach called GOL. It contains two types of geometric constraints that enforce directional (order) and distance (metric) relationships between object instances according to their ranks in an embedding space. Specifically, the order constraint sorts instances directionally according to the ranks, while the metric constraint separates two instances farther if their rank difference is larger. Figure 2 is an overview of GOL.

**Order constraint:**   Suppose that there are $M$ ranks in a training set $\mathcal{X}$. Without loss of generality, the ranks are assumed to be consecutive integers in $\Theta = \{0, 1, \ldots, M - 1\}$. In Figure 2, an encoder $h$ maps each instance $x \in \mathcal{X}$ into a feature vector $h_x = h(x)$ in an embedding space. As $h$, we adopt VGG16 (Simonyan & Zisserman, 2015) without fully connected layers. The output of the last pooling layer is normalized so that $h_x^t h_x = 1$. Thus, the embedding space is a unit hypersphere.

As in Lim et al. (2020) and Lee & Kim (2021), we classify the ordering between two instances $x$ and $y$ in $\mathcal{X}$ into three categories:

$$x \succ y \text{ if } \theta(x) - \theta(y) > \tau, \quad x \approx y \text{ if } |\theta(x) - \theta(y)| \leq \tau, \quad x \prec y \text{ if } \theta(x) - \theta(y) < -\tau, \quad (1)$$

where $\tau$ is a threshold. For instance ordering, notations '$\prec, \approx, \succ$' are used instead of '$<, =, >$.'

The order constraint encourages instances to be sorted according to their ordering relationships. In other words, for two instances $x$ and $y$ with ordering $x \prec y$, the vector from $h_x$ to $h_y$ should be aligned with the direction of the rank increment in the embedding space. To model such rank directions, we introduce $M$ reference points, $r_0, r_1, \ldots, r_{M-1}$, which are learnable parameters guiding the positions of the $M$ ranks in the embedding space. These reference points are randomly initialized by the Glorot normal method (Glorot & Bengio, 2010) and jointly optimized with encoder parameters during training.

Let us define the direction vector $v(r, s)$ from point $r$ to point $s$ on the unit hypersphere as

$$v(r, s) = (s - r)/\|s - r\|. \quad (2)$$

Then, $v(r_i, r_j)$ is called the rank direction from rank $i$ to rank $j$. Also, the rank direction $v(r_i, r_j)$ is forward if $i < j$, and backward if $i > j$. Note that forward directions may differ from one another, for they may represent different physical changes. For example, in age estimation, visual variations from 0-years-old to 5-years-old are mainly due to craniofacial development, whereas those from 45-years-old to 50-years-old are due to skin aging (Geng et al., 2007).

If $x \prec y$, we determine the forward and backward rank directions, respectively, by

$$v_{\mathrm{f}} = v(r_{\theta(x)}, r_{\theta(y)}), \tag{3}$$

$$v_{\mathrm{b}} = v(r_{\theta(x)}, r_{\theta(x)-1}). \tag{4}$$

Then, the encoder is trained so that the embedded features $h_x$ and $h_y$ satisfy the order constraint:

$$x \prec y \quad \Leftrightarrow \quad v_{\mathrm{f}}^t v(h_x, h_y) > v_{\mathrm{b}}^t v(h_x, h_y). \tag{5}$$

In other words, the direction vector $v(h_x, h_y)$ should be aligned more with the forward direction $v_{\mathrm{f}}$ than with the backward direction $v_{\mathrm{b}}$.

To enforce the order constraint in (5), we compute the softmax probability $p^{xy} = [p_{\mathrm{f}}^{xy}, p_{\mathrm{b}}^{xy}]^t$, where $p_{\mathrm{f}}^{xy} = e^{v_{\mathrm{f}}^t v(h_x, h_y)} / (e^{v_{\mathrm{f}}^t v(h_x, h_y)} + e^{v_{\mathrm{b}}^t v(h_x, h_y)})$ and $p_{\mathrm{b}}^{xy} = 1 - p_{\mathrm{f}}^{xy}$. We then define the order loss $L_{\mathrm{order}}$ as the cross entropy between $p^{xy}$ and $q^{xy} = [q_{\mathrm{f}}^{xy}, q_{\mathrm{b}}^{xy}]^t = [1, 0]^t$, given by

$$L_{\mathrm{order}} = q_{\mathrm{f}}^{xy} \log p_{\mathrm{f}}^{xy} + q_{\mathrm{b}}^{xy} \log p_{\mathrm{b}}^{xy}. \tag{6}$$

The order loss for case $x \succ y$ is formulated similarly in a symmetric manner.

**Metric constraint:**  Next, we formulate a metric constraint to make the distance between instances in the embedding space reflect their rank difference. Specifically, it is desirable that

$$|\theta(x) - \theta(y)| > \tau \quad \Leftrightarrow \quad d_{\mathrm{e}}(h_x, h_y) > \gamma \tag{7}$$

where $d_{\mathrm{e}}$ is the Euclidean distance in the embedding space, and $\gamma$ is a margin. Note that if $|\theta(x) - \theta(y)| > \tau$, either $x \prec y$ or $x \succ y$ in (1).

Hence, the metric constraint in (7) is equivalent to

$$x \approx y \quad \Leftrightarrow \quad d_{\mathrm{e}}(h_x, h_y) \leq \gamma \tag{8}$$

From the triangle inequality, we have $|d_{\mathrm{e}}(r_i, h_x) - d_{\mathrm{e}}(r_i, h_y)| \leq d_{\mathrm{e}}(h_x, h_y)$ for every reference point $r_i$. So, if $x \approx y$, we also have $|d_{\mathrm{e}}(r_i, h_x) - d_{\mathrm{e}}(r_i, h_y)| \leq \gamma$. To encourage this inequality, we define a loss $L_{x \approx y}$ as

$$L_{x \approx y} = \sum_{i \in \Theta} \max(|d_{\mathrm{e}}(r_i, h_x) - d_{\mathrm{e}}(r_i, h_y)| - \gamma, 0). \tag{9}$$

On the contrary, if $x \prec y$, it should be that $d_{\mathrm{e}}(h_x, h_y) > \gamma$. Thus, we define another loss

$$L_{x \prec y} = \sum_{i:i \leq \theta(x)} \max(d_{\mathrm{e}}(r_i, h_x) - d_{\mathrm{e}}(r_i, h_y) + \gamma, 0) + \sum_{j:j \geq \theta(y)} \max(d_{\mathrm{e}}(r_j, h_y) - d_{\mathrm{e}}(r_j, h_x) + \gamma, 0). \tag{10}$$

To minimize the first sum, $d_{\mathrm{e}}(r_i, h_x)$ should be reduced, while $d_{\mathrm{e}}(r_i, h_y)$ should be increased. Thus, reference points $r_i, 0 \leq i \leq \theta(x)$, are trained to attract $h_x$ and repel $h_y$, as illustrated in Figure 2. The second sum in (10) is similar. We do not use reference points $r_l, \theta(x) < l < \theta(y)$, which tend to be between $h_x$ and $h_y$ and unhelpful for guiding them. The derivation of $L_{x \prec y}$ in (10) from the metric constraint in (7) is provided in Appendix B. Also, $L_{x \succ y}$ is formulated symmetrically.

Then, we define the metric loss $L_{\mathrm{metric}}$ as

$$L_{\mathrm{metric}} = [x \succ y] \cdot L_{x \succ y} + [x \approx y] \cdot L_{x \approx y} + [x \prec y] \cdot L_{x \prec y} \tag{11}$$

where $[\cdot]$ is the indicator function. Note that, in (9) or (10), we make $h_x$ and $h_y$ attract or repel each other indirectly through reference points $r_i$. This is because the direct attraction or repulsion using $d_{\mathrm{e}}(h_x, h_y)$ may cause $h_x$ and $h_y$ to move in arbitrary directions. Because the reference points are trained also with the order loss in (6), they provide proper directional guidance of attraction or repulsion in (9) or (10). Experimental analysis on $L_{x \prec y}$ in (10) is available in Section 4.4.

**Loss function:**  In addition to the losses for the order and metric constraints, we employ the center loss (Nguyen et al., 2018), which aims at locating each reference point $r_i$ at the center of all instances with rank $i$.

$$L_{\mathrm{center}} = d_{\mathrm{e}}(r_{\theta(x)}, h_x) + d_{\mathrm{e}}(r_{\theta(y)}, h_y). \tag{12}$$

Table 1: Comparison of B2W and DRR scores on the MORPH II, CACD, and Adience datasets.

| Algorithm | MORPH II (setting A) | | | CACD (validation split) | | | Adience | | |
|---|---|---|---|---|---|---|---|---|---|
| | B2W | $DRR_{1.0}$ | $DRR_{0.5}$ | B2W | $DRR_{1.0}$ | $DRR_{0.5}$ | B2W | $DRR_{1.0}$ | $DRR_{0.5}$ |
| ML (Schroff et al., 2015) | 41.20 | 10.06 | 8.94 | 3.90 | 2.89 | 2.55 | 5.12 | 3.21 | 2.98 |
| MV (Pan et al., 2018) | 72.47 | 10.37 | 9.80 | 24.21 | 6.23 | 5.84 | 7.08 | 3.29 | 3.24 |
| OL (Lim et al., 2020) | 43.94 | 12.79 | 11.44 | 8.77 | 5.60 | 4.86 | 13.29 | 7.27 | 6.53 |
| MWR-G (Shin et al., 2022) | 30.42 | 10.66 | 9.28 | 9.99 | 6.00 | 5.24 | 7.84 | 5.20 | 4.71 |
| Proposed GOL | 292.72 | 29.40 | 25.77 | 53.06 | 12.59 | 10.82 | 70.31 | 32.09 | 28.37 |

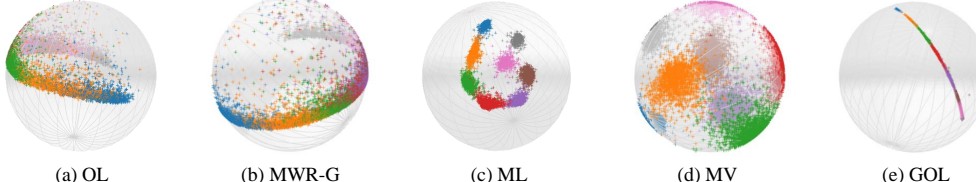

| (a) OL | (b) MWR-G | (c) ML | (d) MV | (e) GOL |

Figure 3: Comparison of embedding spaces on Adience: (a) OL (Lim et al., 2020), (b) MWR-G (Shin et al., 2022), (c) ML (Schroff et al., 2015), (d) MV (Pan et al., 2018), and (e) GOL.

Finally, we use the overall loss function to optimize the encoder parameters and the reference points $r_i$, which is given by

$$L_{\text{total}} = L_{\text{order}} + L_{\text{metric}} + L_{\text{center}}. \tag{13}$$

It is worth pointing out that reference points play essential roles in all three loss terms in (13). They are used to define forward and backward rank directions in $L_{\text{order}}$, so they help to sort instances directionally according to the ranks. The reference points themselves are also sorted since instances are clustered around the reference points because of $L_{\text{center}}$. Also, as mentioned before, the reference points provide proper directional guidance of attraction or repulsion in $L_{\text{metric}}$ to satisfy the metric constraint. In other words, GOL uses the reference points to satisfy both order and metric constraints simultaneously and thus constructs a well-arranged, well-clustered embedding space.

### 3.3 $k$-NN Rank Estimation

For rank estimation, we use the simple $k$-NN rule. Given a test instance $x$, in the embedding space, we find a set $\mathcal{N}$ of its $k$ NNs among all training instances in $\mathcal{X}$. Then, the rank of $x$ is estimated by

$$\hat{\theta}(x) = \frac{1}{k} \sum_{y \in \mathcal{N}} \theta(y). \tag{14}$$

## 4 Experimental Results

We conduct experiments on embedding space construction and rank estimation. Implementation details and more results are available in Appendices C and D, respectively.

### 4.1 Implementation

We initialize an encoder $h$ with VGG16 pre-trained on ILSVRC2012 (Deng et al., 2009) and reference points with the Glorot normal method (Glorot & Bengio, 2010). The Adam optimizer (Kingma & Ba, 2015) is used with a batch size of 32 and a weight decay of $5 \times 10^{-4}$, and the initial learning rates for the encoder and the reference points are set to $10^{-4}$ and $10^{-3}$, respectively. We perform the scheduled learning according to cosine annealing cycles (Huang et al., 2017). For data augmentation, we do random horizontal flips only.

### 4.2 Embedding Spaces

The proposed GOL algorithm attempts to design an embedding space in which both order and metric constraints are satisfied: instances should be well sorted according to their ranks and well separated if they have big rank differences.

Table 2: Comparison of facial age estimation results in the four evaluation settings (A, B, C, and D) of MORPH II. Here, * means that IMDB-WIKI pre-training is performed.

| Algorithm | Setting A | | Setting B | | Setting C | | Setting D | |
|---|---|---|---|---|---|---|---|---|
| | MAE | CS (%) | MAE | CS (%) | MAE | CS (%) | MAE | CS (%) |
| DRFs (Shen et al., 2018) | 2.91 | 82.9 | 2.98 | - | - | - | 2.17 | 91.3 |
| MV (Pan et al., 2018)* | - | - | - | - | 2.79 | - | 2.16 | - |
| C3AE (Chao et al., 2019)* | - | - | - | - | - | - | 2.75 | - |
| BridgeNet (Li et al., 2019)* | 2.38 | 91.0 | 2.63 | 86.0 | - | - | - | - |
| AVDL (Wen et al., 2020)* | 2.37 | - | _2.53_ | - | - | - | **1.94** | - |
| OL (Lim et al., 2020)* | 2.41 | 91.7 | 2.75 | 88.2 | 2.68 | 88.8 | 2.22 | 93.3 |
| DRC-ORID (Lee & Kim, 2021)* | 2.26 | **93.8** | **2.51** | _89.7_ | _2.58_ | _89.5_ | 2.16 | _93.5_ |
| MWR-G (Shin et al., 2022)* | _2.24_ | 93.5 | 2.55 | **90.1** | 2.61 | _89.5_ | 2.16 | 93.0 |
| Proposed GOL | **2.17** | 93.8 | 2.60 | 89.3 | **2.51** | **90.0** | _2.09_ | **94.2** |

To measure the quality of an embedding space for rank estimation, we may adopt the between-class variance to within-class variance (B2W) criterion in Fisher's linear discriminant (Duda et al., 2006),

$$\text{B2W} = \Big( \sum_{i \in \Theta} |\mathcal{X}_i| d_{\mathrm{e}}^2(c_i, c) \Big) / \Big( \sum_{i \in \Theta} \sum_{x \in \mathcal{X}_i} d_{\mathrm{e}}^2(h_x, c_i) \Big) \tag{15}$$

where $\mathcal{X}_i = \{x \in \mathcal{X} \,|\, \theta(x) = i\}$, $c_i = \sum_{x \in \mathcal{X}_i} h_x / |\mathcal{X}_i|$ is the centroid for $\mathcal{X}_i$, and $c = \sum_{x \in \mathcal{X}} h_x / |\mathcal{X}|$ is the centroid for all instances in $\mathcal{X}$. A high B2W score indicates that the rank sets $\mathcal{X}_i, 0 \leq i \leq M-1$, are well separated from one another, while instances in each rank set are compactly distributed.

However, B2W does not consider the ordinal relationships of ranks in $\Theta$ since it was formulated for ordinary classification. Therefore, to assess embedding spaces for rank estimation, we propose the discriminative ratio for ranking (DRR), given by

$$\text{DRR}_\beta = \frac{\sum_{i,j \in \Theta : i<j} (j-i)^\beta d_{\mathrm{e}}(c_i, c_j) / \sum_{i,j \in \Theta : i<j} (j-i)^\beta}{\sum_{i \in \Theta} \sum_{x,y \in \mathcal{X}_i : x \neq y} d_{\mathrm{e}}(h_x, h_y) / \sum_{i \in \Theta} \sum_{x,y \in \mathcal{X}_i : x \neq y} 1} \tag{16}$$

which is the ratio of the average pairwise centroid distance to the average pairwise instance distance in each rank set. Note that, in the numerator, ordinal weights $(j-i)^\beta$ are used to emphasize the difference between a pair of rank sets with a large rank difference. Here, $\beta$ is a nonnegative parameter for controlling the level of emphasis. A high DRR score is obtained when instances are well sorted and well separated according to their ranks in the embedding space.

Table 1 compares the B2W and DRR scores on the MORPH II (Ricanek & Tesafaye, 2006), CACD (Chen et al., 2015), and Adience (Levi & Hassner, 2015) train data. To compare the embedding spaces as fairly as possible, we use the same encoder backbone of VGG16 for all algorithms. GOL significantly outperforms all conventional algorithms in all tests. In GOL, instances within each rank set $\mathcal{X}_i$ are located compactly around reference point $r_i$ by $L_{\mathrm{center}}$ in (12), while those in different rank sets are well separated by $L_{\mathrm{metric}}$ in (11). Hence, GOL yields outstanding B2W scores. Moreover, GOL excels with a larger score gap in $\text{DRR}_{1.0}$ than in $\text{DRR}_{0.5}$, which means that it arranges the rank sets effectively in the embedding space to reflect their ordinal relationships using $L_{\mathrm{order}}$ in (6).

Figure 3 visualizes the embedding spaces on Adience. As in Figure 1, we add a fully connected layer with 3 output neurons to each encoder for the visualization. In (a) and (b), instances are sorted but scattered over the spaces, as the order learning algorithms (Lim et al., 2020; Shin et al., 2022) ignore the metric relation of ranks. In (c), instances in each rank set are well clustered, but the rank sets are not sorted because ML (Schroff et al., 2015) neglects the order relation. In (d), MV (Pan et al., 2018), which is an ordinal regressor, exhibits large within-class scattering, as well as large between-class scattering. In contrast, in (e), GOL constructs a well-sorted, well-clustered embedding space.

### 4.3 Rank Estimation

Since GOL constructs high-quality embedding spaces, it provides excellent rank estimation performances even with the simple $k$-NN rule.

**Facial age estimation:** We use four datasets of MORPH II (Ricanek & Tesafaye, 2006), CACD (Chen et al., 2015), UTK (Zhang et al., 2017), and Adience (Levi & Hassner, 2015), as detailed in Appendix D.3.

Table 3: Comparison in the train and validation settings of CACD and also on UTK.

| Algorithm | Train MAE | Validation MAE | UTK MAE |
|---|---|---|---|
| dLDLF (Shen et al., 2017) | 4.73 | 6.77 | - |
| AGEn (Tan et al., 2017) | 4.68 | - | - |
| DRFs (Shen et al., 2018) | 4.64 | 5.77 | - |
| CORAL (Cao et al., 2020) | - | - | 5.47 |
| Gustafsson et al. (2020) | - | - | 4.65 |
| Berg *et al.* (Berg et al., 2021) | - | - | 4.55 |
| MWR-G (Shin et al., 2022) | 4.76 | 5.75 | 4.49 |
| Proposed GOL | **4.52** | **5.58** | **4.35** |

Table 4: Accuracy (%) and MAE comparison on the Adience and HCI datasets.

| Algorithm | Adience Acc. | Adience MAE | HCI Acc. | HCI MAE |
|---|---|---|---|---|
| OR-CNN (Niu et al., 2016) | 56.7 | 0.54 | 38.7 | 0.95 |
| CNNPOR (Liu et al., 2018) | 57.4 | 0.55 | 50.1 | 0.82 |
| GP-DNNOR (Liu et al., 2019) | 57.4 | 0.54 | 46.6 | 0.76 |
| SORD (Diaz & Marathe, 2019) | 59.6 | 0.49 | - | - |
| DRC-ORID (Lee & Kim, 2021) | - | - | 44.7 | 0.80 |
| POE (Li et al., 2021) | 60.5 | 0.47 | 54.7 | 0.66 |
| MWR-G (Shin et al., 2022) | 62.2 | 0.46 | 52.2 | 0.60 |
| Proposed GOL | **62.5** | **0.43** | **56.2** | **0.55** |

Table 5: Accuracy (%) and MAE comparison on the aesthetics dataset.

| Algorithm | Nature Acc. | Nature MAE | Animal Acc. | Animal MAE | Urban Acc. | Urban MAE | People Acc. | People MAE | Overall Acc. | Overall MAE |
|---|---|---|---|---|---|---|---|---|---|---|
| CNNm (Liu et al., 2018) | 71.0 | 0.31 | 68.0 | 0.34 | 68.2 | 0.36 | 71.6 | 0.32 | 69.5 | 0.33 |
| CNNPOR (Liu et al., 2018) | 71.9 | 0.29 | 69.3 | 0.32 | 69.1 | 0.33 | 69.9 | 0.32 | 70.1 | 0.32 |
| SORD (Diaz & Marathe, 2019) | 73.6 | **0.27** | 70.3 | 0.31 | 73.3 | 0.28 | 70.6 | 0.31 | 72.0 | 0.29 |
| POE (Li et al., 2021) | 73.6 | **0.27** | 71.1 | 0.30 | 72.8 | 0.28 | **72.2** | **0.29** | 72.4 | 0.29 |
| Proposed GOL | **73.8** | **0.27** | **72.4** | **0.28** | **74.2** | **0.26** | 69.6 | 0.31 | **72.7** | **0.28** |

In Table 2, we compare the performances in the four evaluation settings of MORPH II, which is one of the most popular datasets in age estimation. We use the mean absolute error (MAE) and cumulative score (CS) metrics. MAE is the average absolute error between estimated and ground-truth ages (*i.e.* ranks), and CS computes the percentage of images whose absolute errors are less than or equal to a tolerance level $l = 5$. Note that all algorithms in Table 2 use VGG16 as the encoder backbones, except for C3AE employing a shallow CNN. As for Shin et al. (2022), the scores of the global $\rho$-regressor are compared since their local scheme employs as many as six independent VGG16 encoders recursively. GOL performs the best in 5 out of 8 tests, including setting C, which is the most challenging task. Furthermore, unlike most conventional algorithms, GOL does not do the IMDB-WIKI pretraining to boost performances. Even without the pretraining, *i.e.* by employing only MORPH II training data, GOL yields outstanding results in Table 2.

Table 3 compares the results on CACD, which is a bigger dataset containing over 100,000 natural face shots in diverse environments. GOL outperforms the second-best methods with meaningful gaps of 0.12 and 0.17 in the train and validation settings, respectively, which indicates that it can cope with large and diverse data effectively as well.

Table 3 also compares the results on UTK. Note that Gustafsson et al. (2020) and Berg et al. (2021) employ the deep ResNet50 network (He et al., 2016) as their encoders, and MWR-G (Shin et al., 2022) predicts ranks using a complicated, recursive regressor. In contrast, GOL uses the shallower VGG16 encoder and performs rank estimation based on the simple $k$-NN search. Nevertheless, it yields an excellent result, for its geometric constraints help to construct an effective embedding space.

Table 4 shows the results on Adience, where each image is labeled as one of 8 age groups. Compared with the second-best MWR-G, GOL improves the accuracy by 0.3% and reduces MAE by 0.03.

**HCI classification:** The HCI dataset (Palermo et al., 2012) is used for estimating the shooting decade of a photograph. It contains 1,325 images from five decades 1930s $\sim$ 1970s. Table 4 lists the results on HCI. GOL yields the best performances as well, by improving the accuracy by 1.5% and reducing MAE by 0.05 as compared with the second-best methods. This means that GOL yields reliable results even for a small dataset, which may be unfavorable for the $k$-NN estimation.

**Aesthetic score regression:** The aesthetics dataset (Schifanella et al., 2015) contains 15,687 images in four categories. Each image is annotated with a 5-scale aesthetic score. In Table 5, we follow the experimental setting in (Liu et al., 2018). It is challenging to estimate aesthetic scores reliably due to the subjectivity and ambiguity of aesthetic criteria, but GOL performs the best in 8 out of 10 tests. Compared with the state-of-the-art POE (Li et al., 2021), GOL improves the accuracy by 0.3% and reduces MAE by 0.1 overall.

Table 6: Ablation studies for the loss function in (13) on the MORPH II and CACD datasets.

| | $L_{\text{order}}$ | $L_{\text{metric}}$ | $L_{\text{center}}$ | MORPH II (setting A) | | | | CACD (validation split) | | | |
| --- | --- | --- | --- | --- | --- | --- | --- | --- | --- | --- | --- |
| | | | | MAE | B2W | $\text{DRR}_{1.0}$ | $\text{DRR}_{0.5}$ | MAE | B2W | $\text{DRR}_{1.0}$ | $\text{DRR}_{0.5}$ |
| I | | | ✓ | 8.71 | $9 \cdot 10^{-4}$ | 0.07 | 0.06 | 10.0 | 0.05 | 0.42 | 0.38 |
| II | ✓ | | ✓ | 2.71 | 21.48 | 8.89 | 7.69 | 8.59 | 5.16 | 2.66 | 2.65 |
| III | | ✓ | ✓ | 2.43 | 44.94 | 10.93 | 9.63 | 6.14 | 12.72 | 5.79 | 5.07 |
| IV | ✓ | ✓ | ✓ | 2.17 | 292.7 | 29.40 | 25.77 | 5.58 | 53.06 | 12.59 | 10.82 |

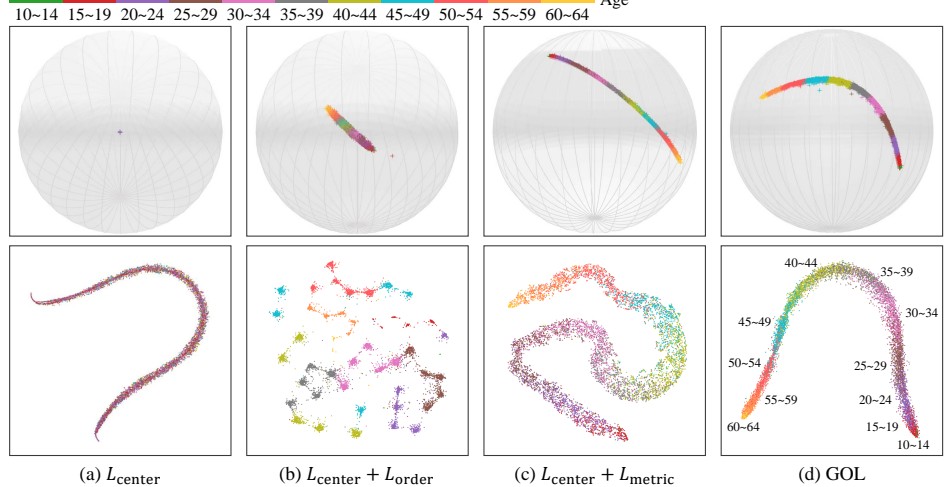

(a) $L_{\text{center}}$     (b) $L_{\text{center}} + L_{\text{order}}$     (c) $L_{\text{center}} + L_{\text{metric}}$     (d) GOL

Figure 4: Visualizing the embedding spaces of the ablated methods in Table 6 on the CACD dataset: (top) reduced 3D spaces and (bottom) t-SNE visualizations of original 512D spaces.

Table 7: MAE comparison of alternative choices for $L_{x \prec y}$ in (10) on MORPH and CACD.

| Method | Alternative to $L_{x \prec y}$ | MORPH (setting A) | CACD |
| --- | --- | --- | --- |
| I | $\max(\gamma - d_{\text{e}}(h_x, h_y), 0)$ | 2.35 | 5.77 |
| II | $\max(d_{\text{e}}(r_{\theta(x)}, h_x) - d_{\text{e}}(r_{\theta(x)}, h_y) + \gamma, 0) + \max(d_{\text{e}}(r_{\theta(y)}, h_y) - d_{\text{e}}(r_{\theta(y)}, h_x) + \gamma, 0)$ | 2.34 | 5.75 |
| III | (10) | 2.17 | 5.58 |

## 4.4 Analysis

**Ablation study:** Table 6 compares ablated methods for the loss function in (13). Method I employs $L_{\text{center}}$ only. In II and III, $L_{\text{metric}}$ and $L_{\text{order}}$ are excluded, respectively. Compared with IV (GOL), method I degrades the performances severely, since the center loss alone cannot construct a meaningful embedding space; a trivial solution to minimize $L_{\text{center}}$ can be obtained by merging all instances and all reference points into a single point. Also, from II and IV, we see that the metric constraint significantly improves the performances by separating instances with different ranks from each other. From III and IV, we see that the order constraint also improves the results by arranging instances directionally according to their ranks. To summarize, both order and metric constraints improve the results and are complementary to each other.

Figure 4 (a)∼(d) show the embedding spaces for the ablated methods I∼IV in Table 6, respectively. The top row visualizes the reduced 3D embedding spaces, as described in Appendix C.2, while the bottom row does the original 512D embedding spaces via t-SNE (Maaten & Hinton, 2008). In (a), only $L_{\text{center}}$ is used, so the 3D embedding space almost collapses to a single point. In (b), instances are directionally aligned, but adjacent rank sets overlap one another. In (c) and (d), the 3D embedding spaces seem similar. However, GOL yields more clearly ordered instances in the t-SNE visualization of the original embedding space than '$L_{\text{center}} + L_{\text{metric}}$' does.

**Alternatives to $L_{x \prec y}$:** Table 7 compares alternative loss terms for $L_{x \prec y}$ in (10). Method I directly increases $d_{\text{e}}(h_x, h_y)$ to make it larger than $\gamma$. Method II uses only two reference points $r_{\theta(x)}$ and $r_{\theta(y)}$. In other words, unlike method III (*i.e.* GOL), it does not employ reference points $r_i$, $0 \le i < \theta(x)$, and $\theta(y) < r_j \le M - 1$. For each method, $L_{x \succ y}$ is also modified accordingly. Method I degrades

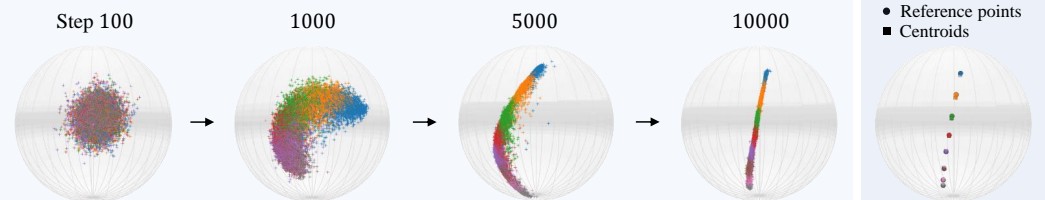

Figure 5: The transition of an embedding space for the Adience dataset during the GOL training.

Table 8: Comparison with order learning algorithms in terms of testing time and network complexity.

| Algorithm | Time (ms) | # parameters (M) |
|---|---|---|
| OL (Lim et al., 2020) | 11.18 | 15.51 |
| DRC-ORID (Lee & Kim, 2021) | 4.93 | 45.86 |
| MWR-G (Shin et al., 2022) | 1.61 | 15.77 |
| Proposed GOL | 0.07 | 14.75 |

the performances badly, as $h_x$ and $h_y$ move in arbitrary directions. Also, GOL performs better than II because attraction and repulsion with many reference points facilitate the positioning of instances in the embedding space, as well as the reference points themselves.

**Embedding space transition:** Figure 5 visualizes the transition of an embedding space for Adience. As the GOL training step goes on, we see that instances are gradually sorted and separated according to their ranks to satisfy the geometric constraints. After the convergence, for each rank, the reference point $r_i$ and the centroid $c_i$ are close to each other, indicating that instances are clustered around their reference points. Moreover, those reference points are well sorted as well.

**Comparison with order learning:** The conventional order learning algorithms (Lim et al., 2020; Lee & Kim, 2021; Shin et al., 2022) provide competitive ranking performances, but they require relatively heavy computations. Table 8 lists the testing times. To estimate the rank of a test instance, OL uses five references for each of the $M$ ranks; *e.g.*, 300 comparisons should be made if $M = 60$ as in typical age estimation. Moreover, OL and DRC-ORID need additional processes, such as MAP estimation, to estimate a rank based on ordering relationships. Also, MWR-G estimates the rank by comparing a test instance with multiple reference pairs recursively. In contrast, GOL simply performs the $k$-NN estimation, which can be done efficiently in a parallel manner, and thus is about 160 and 23 times faster than OL and MWR-G, respectively.

Furthermore, the order learning algorithms should select references among a training set $\mathcal{X}$ through a complicated optimization process; *e.g.*, OL compares all possible pairs of training instances with $O(|\mathcal{X}|^2)$ complexity to select the most reliable references, and DRC-ORID (Lee & Kim, 2021) performs joint network training and clustering for reference selection. On the contrary, GOL requires no such process. Also, in Table 8, the proposed algorithm demands the fewest parameters because it directly estimates the rank of an instance in the embedding space through the $k$-NN search, whereas the order learning algorithms should adopt comparators or regressors tailored for each task.

## 5 Conclusions

The GOL algorithm for rank estimation was proposed in this work. First, we construct an embedding space based on the two geometric constraints, which enforce the direction and distance between instances to represent the order and metric relations between their ranks. Then, we perform the simple $k$-NN search in the embedding space for rank estimation. Extensive experiments on various rank estimation tasks demonstrated that GOL constructs high-quality embedding spaces and yields excellent rank estimation results.

## Acknowledgments

This work was conducted by Center for Applied Research in Artificial Intelligence (CARAI) grant funded by DAPA and ADD (UD190031RD) and supported by the NRF grants funded by the Korea government (MSIT) (No. NRF-2021R1A4A1031864 and No. NRF-2022R1A2B5B03002310).

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
