# OpenReview forum: "Geometric Order Learning for Rank Estimation"
_NeurIPS.cc/2022/Conference — NeurIPS 2022 Accept_

### Official Review · Reviewer_ptgw · 2022-07-11

**Rating:** 6
**Confidence:** 3
**Ethics Flag:** Yes
**Soundness:** 3 good
**Presentation:** 3 good
**Contribution:** 2 fair

**Summary:**

- The paper introduces an algorithm for learning an embedding space using which the rank of an object can be estimated, e.g. age of a person based on face images.
- The embedding space is constrained such that the ordering and distance between training instances are preserved.
- The authors introduce a metric for embedding space evaluation by repurposing the B2W metric (inter-class variance/intra-class variance) to account for the rank ordering of the data embeddings.

**Questions:**

- The order constraint formulation uses $v_b$ which is based on consecutive ranks while the forward uses difference in references corresponding to $x$ and $y$ ranks - Is there a reason for this asymmetry?
- Where does the gain in test runtime come from? I would assume comparing with 5 references/rank would be faster than searching for the k-nearest neighbor from $N$ training examples.
- Also, comparing methods where one uses a compressed representation in the form of references while the other uses the entire training set is a fair setting. The proposed method does learn references during training but seems to ignore these for inference - can the rank estimation be performed with just the references?

Minor:
- How long is the model trained for? Is there a stopping condition?
- Is there any particular reason for choosing the VGG backbone?


**Ethics Review Area:**

["Inappropriate Potential Applications & Impact  (e.g., human rights concerns)"]

**Limitations:**

- The authors show specific failure cases in facial age estimation where the lighting condition, overexposure, and other image perturbations affect the performance of the model - I think this is a general limitation of the dataset and is not specific to the model. A more careful analysis of the proposed method and its limitations might be useful.
- The paper lacks a discussion on why the embedding space has to be ordered - This is crucial for the premise of the method proposed. A method that embeds each rank in distinct spaces will be good at rank estimation (I believe this would explain the performance of other methods compared in the paper).
- The reported results seem to be based on a single run - It would be useful to report results based on multiple random initializations.

**Strengths And Weaknesses:**

- The paper is easy to read. However, parts of the introduction and related work section are repetitive.
- The order and distance properties listed as preliminaries are common knowledge and seem to add no specific detail to the objective formulation in the paper.
- The central idea of the paper is to combine concepts from order learning and metric learning. The paper presents an objective by combining the two constraints.
- The visualization presented in the paper shows that the proposed method learns embeddings that are ordered.
- Ablation study shows that each loss objective added helps improve the performance of the model.

---

> ### Author Response · Authors · 2022-08-01
> **Response to Reviewer ptgw [2/2]**
>
> * **Training time:** We train the encoder until the loss converges. Table below lists the training epochs and time for each dataset. We use an NVIDIA GeForce RTX 3090 GPU in the experiments. These training details have been included in the revision. Please see L541-543 on P16. Also, we will share the training codes.
> \begin{array}{l|c|c|c|c|c|c}
> \hline  \text{Dataset}& \text{MORPH II (A)} & \text{\hspace{0.1cm}CACD (Train)\hspace{0.1cm}} & \text{\hspace{0.3cm}CACD (Val)\hspace{0.3cm}} & \text{\hspace{0.8cm}UTK\hspace{0.8cm}} &  \text{\hspace{0.8cm}HCI\hspace{0.8cm}} & \text{\hspace{0.3cm}Aesthetics\hspace{0.3cm}} \\\
> \hline
> \text{\\# epochs}  & 250 &  10  & 30 &  50 &   150 & 50 \\\
> \hline
> \text{Training Time (hrs)}  & 5&  7  & 1 &  3 &   1 & 3 \\\
> \hline
> \end{array}
>
> * **VGG16 backbone:** We employ VGG16 as the encoder for a fair comparison with existing algorithms. In the main paper, we compare the proposed algorithm with 19 different algorithms in total. Among them, 14 use VGG16, except for CORAL (ResNet34), Gustafsson et al. (ResNet50), Berg et al. (ResNet50), C3AE (shallow CNN), and OR-CNN (shallow CNN). In particular, except for C3AE, all age estimators in Table 2 use VGG16. This has been clarified. Please see L254-255 on P7.
>
> * **More limitations:** Because the proposed algorithm predicts a rank by the $k$-NN search, it suffers where there are insufficient training instances. For example, GOL yields relatively poor results on MORPH II setting B, which consists of 7,000 training samples and 14,000 test samples. Also, most age estimation datasets contain fewer toddler and elder instances. Thus, GOL yields less accurate estimates on such minority classes. These limitations of GOL have been discussed in the revision. Please see L648-653 on P22.
>
> * **Rationale for ordering ranks in an embedding space:** It is almost impossible to design the ideal encoder, perfectly separating each rank in an embedding space, for there is no clear distinction between ranks in practice. Thus, an instance may be erroneously mapped to the region of a wrong rank. Unlike ordinary classification, different errors have different severities in rank estimation: mistaking a young adult as a toddler is severer than mistaking the young adult as a teenager. Ordering in the embedding space can alleviate such severe errors. This rationale has been clarified. Please see L45-48 on P2.
>
> * **Multiple repetitions of experiments:** Below are five repetitive rank estimation results on various datasets. In this test, the networks are trained from scratch five times. Note that the deviations are negligible, and even the worst repetition outperforms the conventional methods on each dataset. We are doing the repetitions for all experiments and will include the results in the camera-ready.
> \begin{array}{l|c|c|c|c|c}
> \hline  \text{Dataset\hspace{0.3cm}}& \text{MORPH II (A) (Fold 0)} & \text{\hspace{1.0cm}CACD (Val)\hspace{1.0cm}} & \text{\hspace{1.7cm}UTK\hspace{1.7cm}} &  \text{\hspace{1.0cm}HCI (Fold 0)\hspace{1.0cm}} & \text{\hspace{0.5cm}Aesthetics (Fold 0)\hspace{0.5cm}} \\\
> \hline
> \text{MAE}  & 2.18±0.023 &  5.61±0.019  & 4.361±0.028 &  0.543±0.021 &   0.288±0.005 \\\
> \hline
> \end{array}
>
> ***
> We have revised our paper to address your comments faithfully and hope that this revision resolve your concerns. If you have any additional concerns, please let us know.
>
> Thank you again for your constructive comments. We do appreciate them.

---

> ### Author Response · Authors · 2022-08-01
> **Response to Reviewer ptgw [1/2]**
>
> Thank you for your positive review and constructive comments. Please find our responses below.
> ***
> * **Repetitive parts:** We have revised the related work section to remove the repetitive parts. Please see L84-86 on P3.
>
> * **Order and metric properties:** We agree that they are common knowledge. We have removed those properties and shortened the descriptions of order and metric. Please see L112-115 on P3.
>
> * **Asymmetric $v_\mathrm{f}$ and $v_\mathrm{b}$:** We may define $v_\mathrm{f}$ and $v_\mathrm{b}$ symmetrically, as you commented. Then, for $x \prec y$, the backward rank direction would be $v_\mathrm{b} = v(r_{\theta(x)}, r_{\theta(x)- (\theta(y)-\theta(x))}) = v(r_{\theta(x)}, r_{2\theta(x)- \theta(y)})$, instead of Eq. (4). During the development of GOL, we tested this design choice as well, but Eq. (4) yielded more stable results. This is because $v(r_{\theta(x)}, r_{2\theta(x)- \theta(y)})$ tends to be less reliable, especially in early training phases before convergence, than $v(r_{\theta(x)},r_{\theta(x)-1})$ between adjacent ranks is when the rank difference between $x$ and $y$ is large. We are doing more thorough experiments to compare alternative choices for the forward and backward rank directions and will include the results in the camera-ready.
>
> * **Gain in testing time:** The conventional order learning algorithms compare a test instance with references using a comparator or a regressor, which consists of multiple layers demanding sequential computations. Moreover, OL and DRC-ORID need additional processes, such as MAP estimation, to estimate a rank based on ordering relationships, and MWR-G adopts iterative estimation. In contrast, the proposed algorithm performs the simple $k$-NN search for rank estimation. Note that the distances to all samples can be computed efficiently in a parallel manner. Thus, as listed below, the proposed algorithm performs fast even with 44,000 samples for the $k$-NN search. Please see L296-300 on P9 and L640-643 on P22.
> \begin{array}{l|c|c|c|c}
> \hline  & \text{MORPH II (A)} & \text{MORPH II (B)} & \text{MORPH II (C)} &  \text{CACD (Val)} \\\
> \hline
> \text{\\# samples}  & 4,394 &  7,000  & 44,000 &  7,600 \\\
> \hline
> \text{Testing time (ms)}  &  0.05  & 0.06  &  0.08 &  0.06 \\\
> \hline
> \end{array}
>
> * **Comparison with reference-based OL methods:** As you pointed out, the proposed GOL uses all instances in a training set for the $k$-NN search. Please note that MWR-G also uses the entire training set, since it also performs the $k$-NN search to obtain an initial estimate. On the other hand, OL and DRC-ORID use a small number of references. However, those references should be selected from the training set via a complicated scheme to exclude unreliable samples and boost the performances.
> As you suggested, GOL may estimate a rank by employing only ‘learned’ reference points (which are different from references in OL and DRC-ORID, ‘selected’ from training sets), but this scheme would yield poor results. It is because our reference points are learned to guide the region of each rank in the embedding space during training, rather than to be used for the $k$-NN search in testing.
> Table below compares the rank estimation performances of the proposed algorithm and OL on setting A of the MORPH II dataset. Even using the entire training dataset $\\cal X$ as references, the performance of OL is not improved meaningfully. On the other hand, with a smaller number of samples for the $k$-NN search, GOL degrades only slightly. In every case, GOL outperforms OL. Thus, we believe that the comparison with the reference-based OL methods is fair, for the training set is a resource fairly available for all algorithms.
> \begin{array}{l|c|c|c|c}
> \hline \text{\\# references or samples} & 275\~\text{(references in OL)} & \hspace{0.6cm}440\~\text{(10\\% of $\\cal X$)}\hspace{0.6cm} & \hspace{0.5cm}2197\~\text{(50\\% of $\\cal X$)}\hspace{0.5cm} &  \hspace{0.6cm}4394\~\text{($\\cal X$)}\hspace{0.6cm} \\\
> \hline
> \text{OL}  & 2.412 & 2.414  & 2.412 & 2.410 \\\
> \hline
> \text{Proposed GOL}  & 2.184 & 2.178  &  2.174 & 2.173 \\\
> \hline
> \end{array}

---

> > ### Comment · Reviewer_ptgw · 2022-08-07
> > **Comments on response [1/2]**
> >
> > Thanks for your response. The updated text and details provided further improve the clarity of the paper.
> >
> > **Asymmetric $v_f, v_b$**: I figured the asymmetry was due to things not working well - performance-wise. The comment was to get an idea of why this would be the case. I'm not sure what you mean by stable results in your response. The reliability argument I think can be made for the forward direction as well, so it's not something specific to the backward direction.
> >
> > **Test time**: This gives an idea of the differences and the gain in test time.
> >
> > **Comparison**:  I'm a little confused by the table result vs text. Is it the case that the performance is subpar with references but does not degrade with the randomly selected subset of the training data (assuming that this is what is reported in the table)?

---

> > > ### Author Response · Authors · 2022-08-08
> > > **Response to Reviewer ptgw's Feedback**
> > >
> > > Thank you for your feedback. We appreciate it greatly.
> > > ***
> > > **Asymmetric $v_\mathrm{f}$ and $v_\mathrm{b}$:**
> > >
> > > > What we meant by 'stable results' is that the loss converged faster and the performance was slightly improved.
> > >
> > > > Please note that, in the order constraint in Eq. (5), the forward direction serves as a positive guide with which $v(h_x, h_y)$ should be aligned, whereas the backward direction does as a negative guide from which $v(h_x, h_y)$ should be pushed away.
> > >
> > > >  As for the forward direction, we haven't considered modeling it as $v(r_{\theta(x)}, r_{\theta(x)+1})$, instead of $v(r_{\theta(x)}, r_{\theta(y)})$ in Eq. (3), because the latter fits the goal of the order constraint in Eq. (5) directly. Furthermore, when using Eq. (3), both reference points, $r_{\theta(y)}$ as well as $r_{\theta(x)}$, can be learned by minimizing $L_\mathrm{order}$. However, as you suggested, we will do experiments with the alternative choice $v_\mathrm{f}=v(r_{\theta(x)}, r_{\theta(x)+1})$ as well.
> > >
> > > \begin{array}{rlcrl}
> > >    &             & \text{O$(r_{\theta(x)})$} &                  &  \\\
> > >    & \diagup &                                       & \diagdown &  \\\
> > > \text{B$(r_{\theta(x)-1})$}  & &  &   & \text{C$(r_{\theta(x)+1})$}  \\\
> > >     | &         &                                       &     & |  \\\
> > >     | &         &                                       &     & |  \\\
> > > \text{A$(r_{2\theta(x)-\theta(y)})$}  &  &  &   &  \text{D$(r_{\theta(y)})$}\\\
> > >  & \diagdown&   & \diagup &   \\\\
> > >  &  & \bullet &   &   \\\
> > > \end{array}
> > >
> > > > As for the backward direction, our initial experiments indicated that Eq. (4) would be a more reliable negative guide than the symmetric choice $v(r_{\theta(x)}, r_{2\theta(x)-\theta(y)})$. This is illustrated intuitively in the figure above, in which reference points are well arranged in an embedding hypersphere. In this case, the direction OA is more similar to the forward direction OD than the direction OB is, so OA could be less effective as a negative guide. Also, when reference points are not as neatly arranged in an early phase of training, the direction OB of a single link would be more reliable than the direction OA determined by a chain composed of multiple links $(r_{\theta(x)}, r_{\theta(x)-1}), (r_{\theta(x)-1}, r_{\theta(x)-2}), \\cdots, (r_{2\theta(x)-\theta(y)+1},  r_{2\theta(x)-\theta(y)})$. This is because the chain is not flattened enough in the early phase, especially when the difference between $\theta(x)$ and  $\theta(y)$ is large (i.e. when there are many links between O and A). Please refer to Figure 4 on P9 and our supplemental video, which visualize embedding spaces during training.
> > >
> > > > We will investigate various combinations of forward and backward directions, including all the aforementioned alternatives, and provide the results in the revised manuscript.
> > >
> > > **Comparison:**
> > > > Please note that, when the OL references are used, the performance of the proposed GOL is degraded due to the reduced number of references (only 275 references). It is not because the OL references are poorer than randomly selected samples for the $k$-NN search. In fact, GOL yields the same MAE of 2.184 also with randomly selected 275 samples.
> > >
> > > > To be clear, we provided the table above to show that the performance of OL is not improved meaningfully even though more references are employed. Hence, the proposed algorithm does not take unfair advantage of more samples for the $k$-NN search. Also, the table shows that GOL consistently outperforms OL, when the same number of samples (or references) are used.
> > > ***
> > > If you have any additional concerns, please let us know. We will do our best to resolve them.
> > >
> > > Thank you again for your feedback.

---

> > > > ### Comment · Reviewer_ptgw · 2022-08-09
> > > > **Comment on response.**
> > > >
> > > > Thanks for the clarifications. I don't have any other questions.

---

> > > > > ### Author Response · Authors · 2022-08-09
> > > > > **Response to Reviewer ptgw's Comment**
> > > > >
> > > > > Thank you again for your constructive and insightful review on our paper. We do appreciate it.

---

### Official Review · Reviewer_3S1v · 2022-07-11

**Rating:** 6
**Confidence:** 4
**Soundness:** 3 good
**Presentation:** 3 good
**Contribution:** 3 good

**Summary:**

This paper introduces geometric order learning (GOL) method for rank estimation by enforcing two geometric constraints: the order constraint and the metric constraint. The order constraint enforces the feature vectors of instances to be arranged according to their ranks, and the metric constraint makes the distance between instances reflect their rank difference. The paper also proposes discriminative ratio ranking metric to assess the quality of embedding spaces for rank estimation. Extensive experiments demonstrate that GOL constructs effective embedding spaces and yields excellent rank estimation performances.

**Questions:**

- The proposed GOP method rely on reference points, it is not clear how to select these reference points and how they affect the performance of GOP.
- As the network architectures of different methods in Table 1 are different, so whether the comparison is fair?

**Limitations:**

Yes

**Strengths And Weaknesses:**

Strengths：
+ A geometric order learning (GOL) method for rank estimation is proposed by enforcing the order constraint and the metric constraint.
+ A discriminative ratio ranking metric is introduced to assess the quality of embedding spaces for rank estimation.
+ Experiments are conducted to demonstrate that GOL constructs effective embedding spaces and yields excellent rank estimation performances.

Weaknesses:
- The proposed GOP method rely on reference points, it is not clear how to select these reference points and how they affect the performance of GOP.
- As the network architectures of different methods in Table 1 are different, so whether the comparison is fair?

---

> ### Author Response · Authors · 2022-08-01
> **Response to Reviewer 3S1v**
>
> Thank you for your positive review and valuable comments. Please find our responses below.
> ***
> * **Selection of reference points:** Please note that we do not select reference points from object instances in a training set. They are learnable parameters to guide the region of each rank in an embedding space, and they are jointly optimized with encoder parameters during training. This has been more clearly described in the revised manuscript. Please see L148-152 on P4.
>
> * **Different network architectures:** As you pointed out, the algorithms in Table 1 have different network structures. However, to compare their embedding spaces as fairly as possible, we use the same encoder backbone of VGG16 for all the algorithms. We have clarified this in the revision. Please see L230-231 on P7.
> ***
> Every attempt has been made to address your comments faithfully in the revised paper. If you have any additional comments, please let us know.
>
> Thank you again for your positive and constructive comments. We do appreciate them.

---

### Official Review · Reviewer_4nRC · 2022-07-11

**Rating:** 7
**Confidence:** 4
**Soundness:** 3 good
**Presentation:** 3 good
**Contribution:** 3 good

**Summary:**

A GOL embedding space represents the direction and distance between objects represent order and metric relations between their ranks, by enforcing two geometric constraints 1) the order/rank constraint  and 2) metric constraint reflects rank difference.

Estimates a test object rank are achieved by kNN, and a metric called discriminative ratio for ranking (DRR) estimates the quality of rank estimation embedding spaces.

Experiments on facial age estimation, historical color image (HCI) classification, and aesthetic score regression demonstrate that GOL constructs effective  embedding spaces and yields rank estimation performances.

**Questions:**

Would be interesting to know the link to traditional ranking theories such as SIFT-Rank.

In Figure 1 c), are we seeing ordered objects arranged in a "band" around a hypersphere?


**Limitations:**

There are no obvious limitations, except demonstration on more datasets & tasks.

**Strengths And Weaknesses:**

It seems GOL is the first attempt to design an embedding space in which the direction and distance between objects represent their order and metric relations. The GOL algorithm performs best in 80% of benchmark datasets for facial age estimation, HCI classification, and aesthetic score regression.

It is not clear how here, the rank and DRR metric are related to traditional (mathematical) methods defining according rank and order geometry, without learning. For example, ranked gradient orientation features are used for head age estimation, classification, with much success, prior to this work.

E.g. please see SIFT-Rank, SIFT being the "scale-invariant feature transform" (the state-of-the-art before GPU-based deep CNN learning), and Rank being the order statistics of the SIFT descriptor (equivalent to uniform orientation sampled gradient filtering).
https://ieeexplore.ieee.org/document/5206849

Toews, M., Wells, W.M. and Zöllei, L., 2012, October. A feature-based developmental model of the infant brain in structural MRI. In International Conference on Medical Image Computing and Computer-Assisted Intervention (pp. 204-211). Springer, Berlin, Heidelberg.

---

> ### Author Response · Authors · 2022-08-01
> **Response to Reviewer 4nRC**
>
> Thank you for your positive review and insightful suggestions. Please find our responses below.
> ***
> * **Relation to traditional features:**  As you suggested, we have cited and discussed the traditional ranking features and methods (SIFT-Rank and Toews et al. in MICCAI 2012). Please see L81-83 on P3.
>
> * **"Band" around a hypersphere:**  Yes, in Figure 1 (c), objects are arranged in a band on the embedding hypersphere.
>
> * **Experiments on more datasets:** We will include experimental results on more datasets, such as EyePacs dataset [1] and AADB dataset [2], in the camera-ready.
>
>    [1] Kaggle. 2015. Diabetic Retinopathy Detection. https://www.kaggle.com/c/diabetic-retinopathy-detection/overview
>
>    [2] Shu Kong, Xiaohui Shen, Zhe Lin, Radomir Mech, and Charless Fowlkes. Photo aesthetics ranking network with attributes and content adaptation. In ECCV, 2016
> ***
> We have made every attempt to address your comments in the revised manuscript and hope that you find this revision satisfactory. If you have additional concerns, please let us know.
>
> Thank you again for your positive comments. We do appreciate them.

---

### Review · Ethics_Reviewer_wdVt · 2022-07-31

**Recommendation:**

I think the paper can continue to be judged on its technical merits without having an ethical concern.

**Ethics Review:**

Although face age estimation is one of the applications of the paper's proposed methodology, it is not the only use of the approach, and isn't exactly tied to that application. Moreover, although face attribute estimation can be used as an instrument of surveillance and oppression, age and other ordinal attributes are not usually the concerning ones: unordered categorical attributes tend to be the ones most problematic.

---

### Review · Ethics_Reviewer_zsLY · 2022-08-05

**Recommendation:**

It is possible to address the ethical concerns in the current version.
Please specify/cite a few relevant bias removing mechanisms.

**Ethical Issues:**

Yes

**Ethics Review:**

The paper proposes a solution for rank estimation and evaluates it on facial images. The proposed algorithm may inherit biases if the training dataset is biased, which may cause discriminatory effects to certain communities

---

### Author Response · Authors · 2022-07-27
**Author response to all reviewers**

We would like to thank all reviewers for their time and positive reviews. We would also extend our thanks to the area chairs.
We are carefully preparing our responses to all suggested comments, and we will upload our response to each question/comment as soon as possible.

---

### Meta-Review · Area_Chair_BYZC · 2022-08-24

**Recommendation:** Accept
**Confidence:** Certain

**Metareview:**

This paper proposes a new approach named geometric order learning (GOL) for rank estimation. Reviewers found that the idea is novel and the paper is well written. The authors have also clearly addressed most questions from reviewers in their responses. Thus, I recommend the acceptance of this paper.

**Award:**

No

---

### Decision · Program_Chairs · 2022-09-14

Accept